# Views from women and maternity care professionals on routine discussion of previous trauma in the perinatal period: A qualitative evidence synthesis

Joanne Cull[1]*, Gill Thomson[1], Soo Downe[1], Michelle Fine[2], Anastasia Topalidou[1]

**1** School of Community Health and Midwifery, University of Central Lancashire, Preston, England, **2** Public Science Project, The Graduate Center, City University of New York, New York, United States of America

* jcull@uclan.ac.uk

## Abstract

### Background

Over a third of pregnant women (around 250,000) each year in the United Kingdom have experienced trauma such as domestic abuse, childhood trauma or sexual assault. These experiences can have a long-term impact on women's mental and physical health. This global qualitative evidence synthesis explores the views of women and maternity care professionals on routine discussion of previous trauma in the perinatal period.

### Methods

Systematic database searches (MEDLINE, EMBASE, CINAHL Plus, APA PsycINFO and Global Index Medicus) were conducted in July 2021 and updated in April 2022. The quality of each study was assessed using the Critical Appraisal Skills Programme. We thematically synthesised the data and assessed confidence in findings using GRADE-CERQual.

### Results

We included 25 papers, from five countries, published between 2001 and 2022. All the studies were conducted in high-income countries; therefore findings cannot be applied to low- or middle-income countries. Confidence in most of the review findings was moderate or high. The findings are presented in six themes. These themes described how women and clinicians felt trauma discussions were valuable and worthwhile, provided there was adequate time and appropriate referral pathways. However, women often found being asked about previous trauma to be unexpected and intrusive, and women with limited English faced additional challenges. Many pregnant women were unaware of the extent of the trauma they have suffered, or its impact on their lives. Before disclosing trauma, women needed to have a trusting relationship with a clinician; even so, some women chose not to share their histories. Hearing trauma disclosures could be distressing for clinicians.

**Data Availability Statement:** This is a systematic review; references for the included papers are given in the paper.

                                                1 / 22

**Funding:** Joanne Cull is funded by a National Institute for Health Research (NIHR) Wellbeing of Women Doctoral Fellowship (grant number NIHR301525). This paper presents independent research funded by the National Institute for Health Research (NIHR) and the charity Wellbeing of Women. The views expressed are those of the authors and not necessarily those of Wellbeing of Women, the NHS, the NIHR or the Department of Health and Social Care. The funders had no role in study design, data collection and analysis, decision to publish, or preparation of the manuscript. https://www.nihr.ac.uk/ https://www.wellbeingofwomen.org.uk/.

**Competing interests:** The authors have declared that no competing interests exist.

## Conclusion

Discussions of previous trauma should be undertaken when women want to have the discussion, when there is time to understand and respond to the needs and concerns of each individual, and when there are effective resources available for follow up if needed. Continuity of carer should be considered a key feature of routine trauma discussion, as many women will not disclose their histories to a stranger. All women should be provided with information about the impact of trauma and how to independently access support in the event of non-disclosures. Care providers need support to carry out these discussions.

## Introduction

Over a third of pregnant women (~250,000) each year in the United Kingdom (UK) have experienced significant trauma such as violence or sexual abuse in childhood or adulthood [1]. Exposure to trauma can have a severe and prolonged impact on mental health, physical health, and health seeking behaviours [2–4]. Internationally, preventing trauma and reducing its impact is a public health priority [5].

Some women who have experienced trauma will have recovered from their experiences at the time of pregnancy, while others begin the pregnancy with unresolved trauma which negatively affects their mental health and the maternal-infant bond [6, 7]. Women who have experienced trauma may find the perinatal period particularly challenging and find that aspects of maternity care, such as clinical procedures, trigger memories of their previous abuse [8].

An increased understanding of the long-term effects of trauma exposure on both mother and baby have led to calls for universal screening within maternity care [9]. The American College of Obstetricians and Gynecologists recommend that women's healthcare providers screen all women for current and past trauma [10]. In the UK, whilst National Institute for Health and Care Excellence guidelines do not address routine enquiry about previous trauma, anecdotally a number of maternity hospitals have introduced it at the initial midwife appointment. However, concerns have also been raised that this could be re-traumatising for women [11], increase unnecessary or unwarranted safeguarding referrals [12], or stigmatise women with a history of adverse events [13].

It is evident that routinely discussing prior trauma with pregnant women requires careful consideration and sensitivity to ensure these conversations create value rather than cause harm. To understand whether routine trauma discussion should be carried out, and if so how it should be done, we aimed to collate and synthesise all relevant qualitative studies to capture women's and maternity care professionals' views of this issue.

We use the term 'routine' to indicate raising the issue of previous trauma with all women accessing maternity services, as opposed to selectively for women who the healthcare professional suspects may have suffered trauma. The term 'routine' in this context does not negate the need for personalised care. We have chosen to use the term 'discussion' rather than 'enquiry' as discussion leaves open the possibility that trauma could be raised in a way that does not involve a direct question.

The review question was 'What are the views of women and maternity care professionals on routine discussion of previous trauma in the perinatal period?'. The study was registered in PROSPERO with the reference number CRD42021247160 [14]. The review was informed by guidance produced by the Cochrane Effective Practice and Organisation of Care group on carrying out and reporting qualitative evidence syntheses [15–17].

## Methods

### Reflexive note

We considered our pre-existing beliefs on routine trauma discussion and how these might influence the design and conduct of the review [18].

JC is a midwife and doctoral student. She was unsure at the outset whether routine trauma discussion was beneficial to women. She believed that for some groups of women, who face discrimination based on their class, race, immigration status or a range of other factors, disclosure of trauma could be harmful and increase the risk of unnecessary or unwanted safeguarding or mental health involvement. SD is a midwife with 18 years of clinical practice and a maternity care researcher, and held similar beliefs to JC. GT has a psychology academic background, and a long history of undertaking research with women who have experienced perinatal mental health problems. Her views were that conversations that were trauma-informed were important to enable needs-led care. AT is a researcher in the field of maternal and neonatal care, and her views were that an appropriate supporting model of care is needed to enable trauma-informed conversations, that will be beneficial to women. MF teaches critical psychology in the US and is a visiting professor in South Africa. With years of participatory work with women in and out of prison, highly marginalized young people and most recently Muslim American youth, MF is interested in collective and individual trauma as sites of wounds and creativity, memory and activism, a rich and painful source of knowledge and wisdom.

A 'research collective' comprising experts by lived experience, from the voluntary sector and healthcare professionals is supporting the doctoral study of which this review is part. Involvement of the collective in the design of the review helped minimise the risk that our pre-understandings would influence the review. Further, the range of backgrounds and views on the review team also helped enhance the rigor of the analysis.

### Search strategy and selection criteria

We searched the databases MEDLINE, CINAHL Plus, EMBASE, APA Psycinfo, and Global Index Medicus using the search terms 'trauma-informed' and 'trauma informed'.

We developed the search strategy by reviewing key relevant papers we were already aware of and examining relevant systematic reviews on PROSPERO. Each of the key papers referred to 'trauma-informed' in their keywords, title or abstract. We trialled searching using this term on Medline and duplicated it using a variety of different terms such as trauma-sensitive, trauma-focused, and trauma-responsive. These alternative terms had very few hits and had always been used in addition to trauma-informed. We therefore decided to only use the search terms 'trauma-informed' or 'trauma informed' and hand select relevant papers that concerned the care of women in the perinatal period.

Forward and backward citation tracking and key author searches for studies included in the review were carried out to identify additional relevant studies. Searches were conducted in July 2021 and updated in April 2022.

Only qualitative studies and qualitative aspects of mixed methods studies were included in the review. Studies not based in a maternity setting, or which did not include women in the perinatal period (defined for this purpose as pregnancy and up to one year after birth) were excluded. The review was focused on previous psychological trauma: this could include all past trauma or specific types such as adverse childhood experiences or sexual abuse but excluded studies with participants who have experienced physical trauma such as injury. Studies published at any date, and in any language were eligible for inclusion in the review.

Two reviewers (GT and JC) independently screened 20% of papers at title and abstract stage using Rayyan blind screening. The level of agreement was set at 95%. As the reviewers

achieved 100% agreement, the remaining 80% of papers were reviewed by JC. At the second stage, all papers were blind screened by two reviewers (GT and JC), with any differences of opinion about inclusion resolved through discussion.

## Data extraction and management

Papers eligible for inclusion were uploaded to the software program MAXQDA Plus 2020 for data extraction, analysis, and thematic synthesis. We developed a standardised data extraction form using Excel and piloted it prior to beginning data collection. JC used the form to record basic contextual and methodological information about each study, including bibliographic information, country of study, setting, study design, data collection, participants' characteristics, data analysis methods and key themes. To reduce bias and errors, GT independently extracted data from 20% of studies, and JC and GT compared results, resolving disagreements through dialogue.

## Appraisal of the methodological quality of included studies

JC assessed the quality of each included paper using the Critical Appraisal Skills Programme (CASP) quality assessment checklist for qualitative studies [19]. The CASP checklist comprises 10 questions: 1 mark was allocated to each question if the criterion was met. The overall quality of each study was categorised as 'strong' (score 8-10/10; minimal methodological issues), 'adequate' (score 5-7/10; no major methodological issues) or 'weak' (0-4/10; major methodological issues). Studies scoring 'weak' were excluded on quality grounds to ensure that the credibility or trustworthiness of the review findings was not compromised by including studies with important methodological limitations [15]. GT independently quality assessed 20% of included studies, and JC and GT compared results, resolving disagreement through re-examination and discussion. As suggested by Carroll and Booth [20] post hoc sensitivity analyses were carried out to assess the impact on the review findings of excluding a study on quality grounds. Quality ratings contributed to the GRADE-CERQual assessments (described below).

When we were quality appraising each study, we also reflected on broader issues of the integration and inclusion of women's voices, and these additional considerations are reflected in the discussion.

## Data synthesis

Data were synthesised thematically using the method developed by Thomas and Harden [21]. First, the findings of each study were inductively coded on a line-by-line basis. The codes were then organised into related areas, constructing 'descriptive' themes (summaries of findings). Finally, the descriptive themes were organised into analytical themes. This work was led by JC, and, at all stages, emerging concepts were shared, discussed, and refined with the review team.

## Assessment of confidence in the review findings

The GRADE 'Confidence in the Evidence from Reviews of Qualitative research' (GRADE-CERQual) approach was used to assess confidence in the synthesised findings [22–27]. This approach facilitates explicit and transparent assessment of whether the findings from a qualitative synthesis reasonably represent the phenomenon of interest. Each summary of findings (descriptive theme) was assessed in terms of methodological limitations, coherence, adequacy of data, and relevance; with the four assessments contributing to an overall assessment of confidence. The summaries of findings and associated CERQual assessment of confidence are presented in an Evidence Profile and Summary of Qualitative Findings table.

## Results of the search

Overall, 3,888 papers were identified after removal of duplicates, of which 25 met the inclusion criteria for the study. The PRISMA flow chart can be found at Fig 1. Three papers were excluded due to quality [28–30]. Post hoc examination of the three excluded papers indicated that their inclusion in the review would not have altered the final themes.

## Description of the studies

The characteristics and quality of the 25 studies included in the review are summarised in Table 1. The included studies were published between 2001 and 2022. In terms of study setting, 12 of the studies were carried out in Australia, nine in the United States, two in Sweden, and one each in England and Canada. Routine trauma discussion was explored from the perspective of women in thirteen of the papers, eight looked at the perspective of healthcare professionals, and the remaining four papers looked at both viewpoints. Most data were collected by individual interviews, focus groups and/or surveys. The studies represented the views of 1602 women and 286 healthcare professionals and experts from the voluntary sector.

## Assessment of methodological strengths and limitations

Of the 25 included papers, sixteen were assessed as methodologically strong and nine as adequate. As discussed above, a further three papers were assessed as methodologically weak and were excluded from the review. A lack of reflexivity was noted across most of the studies. Other common methodological weaknesses included insufficient information about data analysis, the lack of a clear statement of findings, and minimal discussion of ethical issues.

## Confidence in the review findings

Table 2 shows the Summary of Findings and CERQual rating for each summary of finding. Confidence in most of the review findings was moderate or high, reflecting the quality and quantity of the studies included in the review. Each summary of finding was mapped to an analytical theme and these themes are discussed in the next section. All studies that met the inclusion criteria were from high-income countries. Because these findings were applicable to the UK, we did not downgrade the CERQual ratings for income status of the countries. However, the findings are not applicable to low- and middle- income countries.

# Findings

Six analytical themes were identified relating to women's and maternity care providers' views and experiences of routine trauma discussion. The first theme *'I did not know how to say it, and no-one asked me'* considers whether maternity care providers should ask women about previous trauma. *'A real whitefella way to start'* explores standardised compared to more unstructured ways of asking about trauma. In the theme *'You say it is confidential. . . but you are going to report me'*, fear of judgement as a barrier to disclosure, and the importance of trust and relationships in trauma discussions is highlighted. The theme *'I'm not quite sure what is going on, but I feel really vulnerable'* calls attention to the intensity of the perinatal period, which is often challenging but also has the potential for healing and growth. *Embedding trauma in routine practice* explores barriers and facilitators to successful implementation. Finally, *'You go home and it's playing on your mind as you're cooking'* considers the impact on care providers of hearing trauma disclosures.

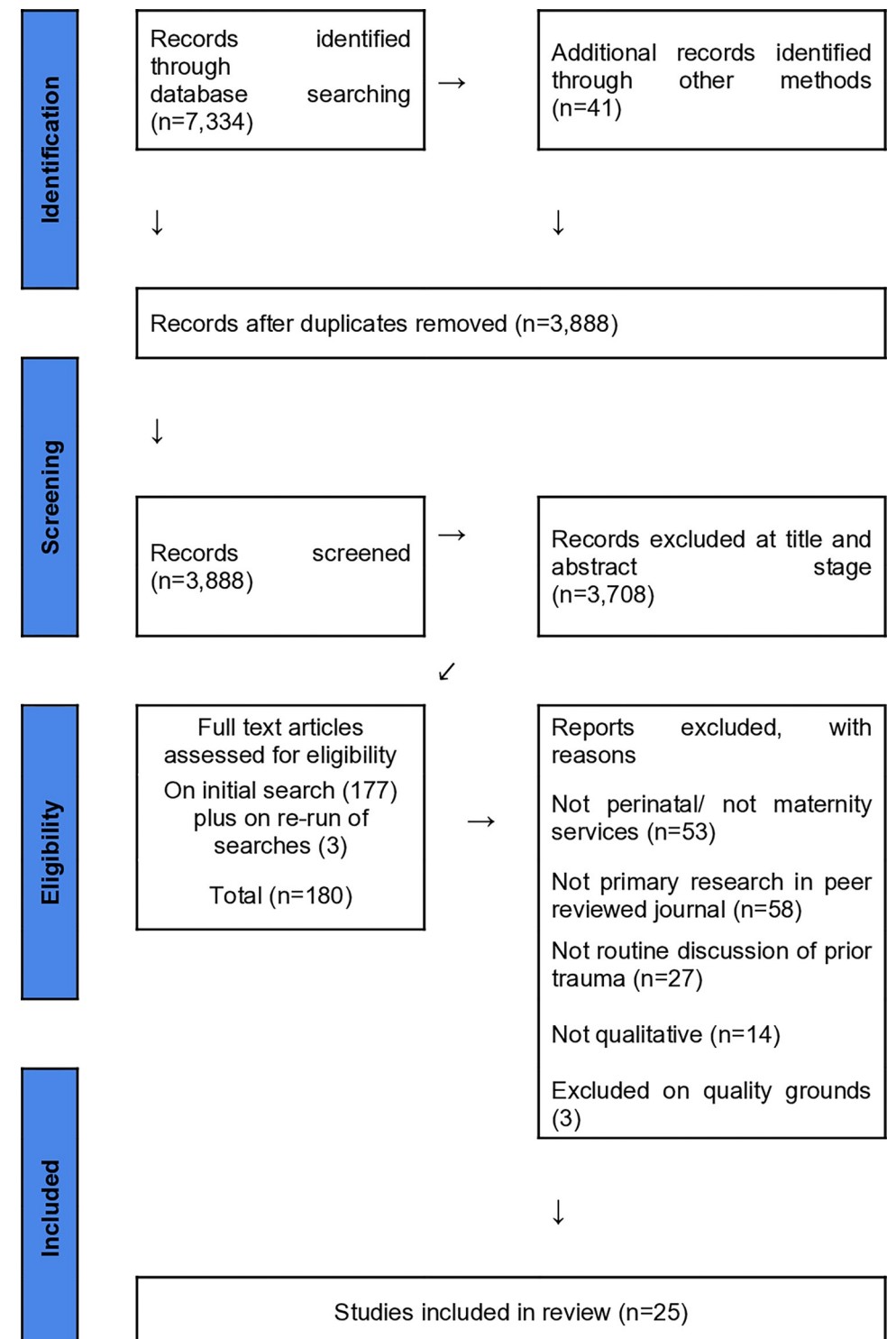

**Fig 1. PRISMA flow chart.**

**Table 1. Characteristics of included studies.**

| Study no. | Authors | Year | Country | Study design | Participants (number / type) | CASP Quality assessment rating | Focus of paper |
|---|---|---|---|---|---|---|---|
| 1 | Carlin, Atkinson and Marley | 2019 | Australia | Yarning—conversational process involving telling of stories and development of knowledge | 15 Aboriginal women | Strong | Women's perspectives |
| 2 | Carlin et al. | 2020 | Australia | Health professionals—online survey or semi-structured interview. Aboriginal women—in depth interviews | 18 health professionals 10 Aboriginal women | Strong | Women and clinician perspectives |
| 3 | Chamberlain et al. | 2020 | Australia | Stakeholder workshop | 57 key stakeholders, with extensive experience working with Aboriginal families. | Strong | Women and clinician perspectives |
| 4 | Choi and Seng | 2014 | United States | Semi-structured telephone interviews | 20 perinatal care providers | Strong | Clinician perspectives |
| 5 | Flanagan et al. | 2018 | United States | Childbearing women—ACE questionnaire plus telephone interview. Clinicians—surveys and focus groups | 210 childbearing women; 26 clinicians | Adequate | Women and clinician perspectives |
| 6 | Gokhale et al. | 2020 | United States | Semi-structured interviews and completion of trauma history questionnaire | 30 pregnant women | Strong | Women's perspectives |
| 7 | Kohlkoff et al. | 2021 | Australia | Focus groups and semi-structured interviews | Nine midwives, two obstetricians, and one nephrologist | Adequate | Clinician perspectives |
| 8 | Marley et al. | 2017 | Australia | Childbearing women—questionnaire. Clinicians—questionnaire and follow-up interview | 81 women; 9 study personnel | Adequate | Women and clinician perspectives |
| 9 | Mendel, Sperlich, and Fava | 2021 | United States | Semi structured interviews | 99 first time mothers | Adequate | Women's perspectives |
| 10 | Millar et al. | 2021 | Canada | Questionnaire consisting of ACE-10 plus open- and closed-ended questions; semi-structured interviews | Questionnaire—29 adolescent mothers Follow- up interview– 5 mothers | Strong | Women's perspectives |
| 11 | Mollart, Newing and Foureur | 2009 | Australia | Focus group interviews | 18 midwives from 2 study sites | Strong | Clinician perspectives |
| 12 | Montgomery, Seng and Chang | 2021 | England | Focus groups, interviews, and an online survey | 2 focus groups, 2 interviews, 29 responses to online survey | Strong | Women's perspectives |
| 13 | Mule et al. | 2021 | Australia | Open-ended question giving reason for non-disclosure of trauma history | 161 childbearing women | Adequate | Women's perspectives |
| 14 | Olsen, Galloway and Guthman | 2021 | United States | Online survey with quantitative and open-ended questions | 154 women | Strong | Women's perspectives |
| 15 | Reilly et al. | 2020 | Australia | Semi-structured interviews | 3 midwives, 3 obstetricians, 2 managers, 1 mental health worker | Strong | Clinician perspectives |
| 16 | Rollans et al. | 2013 | Australia | Qualitative ethnographic study–observation of antenatal and postnatal appointments plus face to face interviews. | 34 observed antenatally; 20 of the same women who were observed during postnatal visit; 31 antenatal interviews, 29 postnatal interviews | Strong | Women's perspectives |
| 17 | Schmied et al. | 2020 | Australia | Survey before introduction of new psychosocial assessment, second survey following implementation; focus groups | First survey—26 midwives, second survey—27 midwives (9 midwives completed both). Focus groups—16 midwives | Strong | Clinician perspectives |
| 18 | Seng et al. | 2002 | United States | Narrative interviews | 15 childbearing women | Strong | Women's perspectives |
| 19 | Sobel et al. | 2018 | United States | Semi-structured interviews | 20 women with history of sexual trauma; 10 without | Adequate | Women's perspectives |
| 20 | Stenson et al. | 2001 | Sweden | Open-ended written / telephone question about abuse screening in pregnancy | 879 women | Adequate | Women's perspectives |

*(Continued)*

**Table 1.** (Continued)

| Study no. | Authors | Year | Country | Study design | Participants (number / type) | CASP Quality assessment rating | Focus of paper |
|-----------|---------|------|---------|--------------|------------------------------|--------------------------------|----------------|
| 21 | Stenson, Sidenvall and Heimer | 2005 | Sweden | Focus groups | 21 midwives in 5 focus groups | Adequate | Clinician perspectives |
| 22 | White, Danis and Gillece | 2015 | United States | Focus group | 6 women | Strong | Women's perspectives |
| 23 | Willey et al. | 2020a | Australia | Focus groups and semi-structured interviews | 24 healthcare professionals | Strong | Clinician perspectives |
| 24 | Willey et al. | 2020b | Australia | Focus group and semi-structured interviews | 22 women who were refugees, 5 women who were migrants | Strong | Women's perspectives |
| 25 | Preis et al. | 2022 | United States | Focus groups and semi-structured interviews | 22 healthcare professionals | Adequate | Clinician's perspectives |

### 'I did not know how to say it, and no-one asked me': Should maternity care providers ask women about previous trauma?

This theme explores whether women and maternity care providers feel that routine trauma discussion should take place.

Participants in 14 studies expressed that they felt routine trauma discussion is acceptable and worthwhile [31–44]. One of the participants in the study undertaken by Montgomery, Seng and Chang (2021) [36] proposed:

> "It might have just put a thought in my head, even if it wasn't something that I shared with anybody, it might have just put a thought in my head which might have been useful at some point" [36].

Some women felt it was difficult to broach the subject of previous trauma and would not have disclosed unless the clinician raised the issue:

> "At the time, I could not and did not tell the healthcare professionals of my survivor status. I did not know how to say it, and no one asked me" [36].

The overall finding that women accepted routine trauma discussion masks several complexities and contradictions. In eight studies, women reported feeling unprepared for the discussion and found it intrusive [32, 33, 35–39, 42]. This was reflected by a woman who was interviewed in the study by Millar et al. (2021) [35]:

> "Like it doesn't feel good when you first meet someone, and they just start like trying to jump into your life. Like they know you. I hate that" [35].

Some participants spoke of how they would have welcomed forewarning of the discussion:

> "I think they could have told me what they were going to ask before I even arrived for my appointment. I had no idea that was what was coming" [39].

In 11 studies, professionals reported that they felt routine trauma discussion was worthwhile [32, 34, 45–53]. A clinician taking part in the study by Flanagan et al. (2018) [47]

**Table 2. Summary of findings and CERQual ratings.**

| Review finding | Relevant studies (study numbers as per Table 1) | CERQual assessment of confidence in the evidence | Theme |
|---|---|---|---|
| Women feel positively about routine trauma discussion | 14 studies (1,2,6,8,10,12,13,14,16,18,19,20,22,24) | Low | 'I did not know how to say it, and no-one asked me': should maternity care providers ask women about previous trauma? |
| Some women find routine trauma discussion invasive and unexpected | 8 studies (2,6,13,14,16, 20) | High | |
| Maternity care providers feel routine trauma discussion is valuable | 11 studies (2,3,4,5,7,8,15,17,21,23,25) | Moderate | |
| Support for routine trauma discussion is contingent on adequate time and resources | 16 studies (2,3,4,5,7,8,14,15,16,17,19,20,21,22,23,24) | High | |
| Women favour a broad, conversational approach to discussing trauma | 5 studies (1,2,9,12,13) | Very low | 'A real whitefella way to start': standardisation and tickboxes in trauma discussion |
| Women who have suffered trauma want relationship-based care | 6 studies (1,6,10,12,14,18) | Moderate | |
| Choice and control is important to women | 6 studies (1,10,12,14,18,19) | High | |
| Women want further therapeutic support | 7 studies (1,3,6,10,14,22,24) | Moderate | |
| Women fear judgement if they disclose their histories | 12 studies (1,2,3,12,13,14,17,18,19,20,22,24) | High | 'You say it is confidential. . . but you are going to report me': the importance of trust |
| Relationships are a critical prerequisite to trauma disclosure | 13 studies (1,2,3,6,10,12,13,14,16,18,20,21,24) | High | |
| The manner of the person asking and the environment are also important | 12 studies (1,2,3,6,8,13,14,16,18,19,22,24) | High | |
| If not handled sensitively, trauma discussion could affect future health care access and experiences | 3 studies (12,14,16) | Low | |
| Some women will choose not to disclose previous trauma | 13 studies (2,3,4,6,10,12,13,14,16,17,18,19,24) | High | |
| Some women feel their previous experiences are irrelevant to their current pregnancy | 7 studies (3,6,12,13,16,18,20) | Moderate | 'I'm not quite sure what is going on, but I feel really vulnerable': the intensity of the perinatal period |
| The perinatal period can be intense and challenging | 10 studies (1,2,4,10,12,14,18,19,20,24) | High | |
| Not all women were fully aware of the extent or impact of the trauma they had suffered | 10 studies (1,2,3,4,6,12,13,14,18,22) | High | |
| The perinatal period carries potential for healing and growth | 7 studies (1,4,6,8,12,18,19) | Moderate | |
| Embedding trauma discussion in routine practice is challenging | 7 studies (4,5,7,15,17,21,23) | Moderate | Embedding trauma discussion in routine practice |
| Partner presence can be a barrier to trauma discussion | 2 studies (7,21) | Very low | |
| Women with limited English face additional challenges in discussing trauma | 3 studies (16,23,24) | Low | |
| Hearing trauma disclosures can be distressing for maternity care providers | 5 studies (4,7,8,11,21) | Low | 'You go home and it's playing on your mind as you're cooking': the impact on care providers of hearing trauma disclosures |

reflected that while '*most of the time (the) screen is negative*', when finding the individual who had faced previous trauma, '*you're so glad you did*' [47].

Participants in the study by Kohlkoff et al. (2021) felt there were three key benefits of routine trauma discussion: identify women at higher risk of mental health problems or family

violence, increase referrals to appropriate support services, and provide support and education [47]. However, in seven studies, women felt that trauma discussions should only take place if clinicians had enough time to respond to disclosures and could provide or refer into appropriate support [32, 38, 39, 41–44]. Having disclosed prior trauma, some women expected that they would be treated more sensitively and that other care providers would be aware of their history, and were aggravated when this wasn't the case [39, 41]:

> "*Why don't they take the extra time just to read over [my file] and if they have any more questions about it then they can ask. If it's already there then why bother. . . it is really frustrating*" [39].

Maternity care providers similarly spoke of their support for routine trauma discussion as contingent on having sufficient time and appropriate referral pathways [32, 34, 45–48, 50–53]. Without good quality support services, clinicians were reluctant to discuss prior trauma, fearing this would open a 'Pandora's box' of issues they were unable to deal with:

> "*We see perinatal depression and anxiety but this is a continuum of social disadvantage and intergenerational trauma. We have super complicated patients with so many problems. Where do we fit mental health in where there are so few resources to respond properly?*" [32].

### *'A real whitefella way to start'*: Standardisation and tick-boxes in trauma discussion

This theme explores how maternity care providers should raise the issue of previous trauma. Participants in several studies discussed limitations in the use of questionnaires to raise the issue of previous trauma [31, 32, 37, 45, 54]. Participants suggested that closed questions (for example 'in the last year, have you experienced. . .') can prevent disclosures [31, 37] This Aboriginal participant in a study by Carlin, Atkinson and Marley (2020) [32] proposed:

> "*You talk about things because they are important to talk about not cause they happened one week ago! It is a real whitefella way to start. It's like you're in or you're out. You see that hey? Like what happens if it was a bit longer, then the lady might think oh no, it's not important, I won't talk about that*" [31].

Mendel, Sperlich and Fava (2021) [54] investigated the use of the Adverse Childhood Experiences questionnaire (ACE-10) within maternity research. The researchers found that the questionnaire contains confusing and ambiguous questions, excludes important traumatic events in childhood (such as the death of a parent) and fails to ascertain the severity or duration of the traumatic experience. For example, one participant in the study had an ACE score of one out of ten but had suffered extensive abuse over 12 years of her childhood, resulting in seven miscarriages. The authors concluded that completion of the questionnaire might not give a true representation of the extent of trauma the woman has suffered.

Women who took part in the study by Carlin et al. (2020) [32] felt that direct questions could cause women to disengage, and that broad, gentle questions were the best approach to ask about difficult experiences. The voluntary sector experts and healthcare professionals who participated in the study by Chamberlain et al. (2020) [45] noted that direct questions about trauma can be problematic, because avoiding thinking about trauma experiences can be a way of coping. Instead, indirect methods of gentle communication were preferred, asked by *"someone trusted–this sort of information will naturally become evident so the trusted person can*

*gently empathise and draw attention to, as opposed to ask directly and abruptly"* [45]. Another participant noted that clinicians should ask *'slowly, gently and only where there is the possibility of being able to 'hold a space' and deal appropriately with the answer'* [45].

Carlin et al. (2020) [32] explored clinicians' views of a questionnaire-based approach versus questionnaire plus narrative. Several participants reported that they only used the questionnaire, describing the narrative section as 'aspirational' due to time constraints and concerns that it positioned them as a counsellor. However, those who did use the narrative approach felt it enhanced rapport with women and that women understood the limits around the assistance they could provide:

"*Generally I think women are keen to share some of their problems with us as nurses even though we cannot solve these issues as such but we can listen, we can advise them where to seek help and how we can assist as a support for some of their problems.*" [32].

## 'You say it is confidential. . . but you are going to report me': The importance of trust

This theme concerns the importance of trust and relationships to women who have suffered trauma. In twelve studies, fear of judgement was reported as a reason for non-disclosure [31, 32, 36–38, 40–45, 51]. This included a general fear of being perceived as a bad parent, and specific concerns that their child would be removed from their care; *"You guys are bound by law [to report certain things]. . . You say it is confidential. . . but you are going to report me"* [43]. Concerns about confidentiality were raised by participants in several studies, as was the misconception that the abused becomes the abuser. One mother stated:

"*Speaking from personal experience, I felt at times that my past trauma was being used to assess the likelihood I would harm my own child, rather than as a means of identifying what support I might need as an individual*" [38].

Participants in 13 studies highlighted the importance of a trusting relationship, built through multiple encounters, as a prerequisite for trauma disclosure [31–33, 35–40, 42, 44, 45, 52] A mother from the study by Millar et al. undertaken in 2021 [35] suggested:

"*I think if I had a relationship, then yes I [would disclose trauma history]. But with the amount of time I was seeing them, no. 'Cause I was always seeing someone different*" [35].

Participants in five studies proposed that trauma should not be discussed at the first appointment, but at a later appointment, enabling a relationship to be built first [35, 38, 42, 45, 52]. This woman chose to disclose later in pregnancy:

"*When she gave me the initial, you know, the history form. . . when I saw 'were you abused?' I said no. There was no way I was going to tell her*" [40].

Although maternity care providers agreed that women were more likely to disclose prior trauma after a relationship with the clinician had been established, some felt it was appropriate to raise trauma at the first consultation, seeing this as the start of an ongoing conversation:

"*If. . . they're not opening up. . . they'll go home and think about it and reflect on that and they may come back the next time and open up a bit more. It's just opening the door, isn't it?*" [51].

In twelve studies, participants described the importance of how the clinician asked about previous trauma [31–34, 37–41, 43–45]. Desired attributes consistently included kindness, friendliness, sensitivity, a non-judgemental attitude, respect, care, and compassion. One woman who took part in the study carried out by Sobel et al. (2018) [41] expressed that *"I opened up to my midwife because I felt comfortable with her. That's it"* [41] Conversely, this participant in Seng et al.'s study (2002) [40] felt the need to change care provider:

*"The doctor was kind of cold, not personable at all, and those feelings [emotional memory of being abused, shame, vulnerability, nakedness] would come back to me in his office, and I found myself crying at every visit"* [40].

For women with limited English, non-verbal signals like smiling and a relaxed manner were vital in inspiring trust: *"She make me like not scared because she smile a lot, her smiling and the way she spoke was really helpful"* [39].

Participants proposed that trauma discussions should be held in private, comfortable and welcoming surroundings [33, 38, 43, 45]. The ideal combination of a trusted care provider and warm environment is summed up by a woman who took part in Gokhale et al.'s study (2020) [33]:

*"The only way my health care providers can help me with my trauma is every time I come, make me feel like I'm at home. Make me feel comfortable. Make me feel safe and make me feel like I have nothing to worry about".*

Participants in three studies felt that if the conversation was handled badly, routine trauma discussion could impact upon future health care access and experiences [36, 38, 39]. One respondent to the survey carried out by Olsen, Galloway and Guthman (2021) [38] proposed:

*"If it isn't asked about in a sensitive way under the right circumstances, it could feel really intrusive or could be so upsetting or off-putting that someone could avoid needed health care entirely"* [38]

Participants in thirteen studies reported that they would not disclose previous trauma to the healthcare professional looking after them, although not all studies explored the reasons for this [32, 33, 35–41, 44–46, 51]. Mule et al. (2021) carried out a survey asking women whether they had chosen to fully disclose their histories during antenatal psychosocial assessment: 161 women responded that they had not, and completed an open-ended question giving their reasons. The researchers found there were a range of reasons, including lack of trust of the person asking, fear of judgement, use of closed-ended questions and lack of time, but also simply privacy: they did not want to share this information [37]. Similarly, a participant in the study by Olsen, Galloway and Guthman (2021) proposed that *"some people may not be ready"* [38] while a woman taking part in the study by Gokhale et al. (2020) suggested that *"it's not easy speaking up about situations like that and a lot of people don't because they don't feel comfortable or they don't feel like they could trust people enough to do that"* [33]. This implies that even within a trusting relationship, some women who would benefit from support will choose not to share their histories. Accordingly, participants in the study by Seng et al. (2002) [40] proposed that care providers should assume women are trauma survivors if they display signs or symptoms of trauma, irrespective of whether they have disclosed.

### 'I'm not quite sure what is going on, but I feel really vulnerable': The intensity of the perinatal period

This theme explores experiences of the perinatal period for women who have suffered trauma. In seven of the studies, some participants reported feeling that there was no connection between their trauma histories and their current wellbeing and pregnancy [33, 36, 37, 39, 40, 42, 45]:

> "*It's not really affecting me now. . . my main concern is getting through the pregnancy, not worrying about my past stuff*" [39].

Some women wanted to focus on the pregnancy and a positive future and felt discussion of trauma could trigger distressing feelings. This was the case even when the perinatal period could be expected to bring up strong emotions, such as for this woman whose infant had been murdered:

> "*Cause when you come to the visit you want to hear stuff about your baby. You don't want to keep dwelling on this that happened in the past and you trying to have a happy moment.*" [33].

Some women had not foreseen that their pregnancy would be so difficult: *"It's hard to put into words because I'm not quite sure what is going on, but I feel really super vulnerable"* [40]. Even women who appeared to be far along in recovery and living happy lives were often unprepared for the intensity of the perinatal period:

> "*I was really looking forward to the cuddling time with the baby and breastfeeding. . . I didn't expect this whole other ugliness*" [40].

Women commonly felt a loss of control over their body, due both to the pregnancy and a sense of powerlessness within maternity care. Vaginal examinations, birth, or even seemingly benign clinical procedures such as blood pressure measurement caused flashbacks to abuse. Some women feared bodily exposure during labour and birth, with a participant in the study by Sobel et al. (2018) [41] reporting:

> "*I was so concerned with being covered up. . . I would have been devastated [by a vaginal delivery]. I did not know how I was going to keep my clothes on and have a baby*" [41]

Until pregnancy, some women were not fully aware of the trauma they had suffered. Seng et al. (2002) [40] explored this issue in detail through narrative interviews with 15 women who had suffered childhood sexual trauma and subsequently accessed maternity care. At the time of the pregnancy, four of the fifteen women had only a vague understanding that they had been subjected to abuse. Participants described indications in their thoughts and behaviour of the effects of trauma, such as fleeting flashbacks, suicidal intentions, and extreme promiscuity:

> "*I realized [later] there were pieces that had been floating around for a long time that I wouldn't acknowledge prior to [postpartum]*" [40].

Because they had not fully admitted it to themselves, these women were not able to disclose the abuse to their healthcare providers:

*"I knew early that I was not going to deliver vaginally. I knew in my head that I was not going there. So that piece I connected. . .I don't know that I drew a real direct line because of how vulnerable I felt. I wasn't probably ready to acknowledge that. . .So it was knowing and not knowing at the same time"* [40].

Participants talked about the potential for post-traumatic growth in the perinatal period [31, 40, 41, 46]. However, this was not always an easy process, as described by this participant in the study by Seng et al. (2002) [40]:

*"I kind of knew in some way it was affecting me, but I just couldn't connect the dots ever. . . but when I got pregnant it all just came out, came clear, and it was hard, and I'm grateful. . . and I think it's going to help me grow past it and deal with it. . . but pregnancy is enough to deal with."* [40].

### Embedding trauma discussion in routine practice

This theme investigates how trauma discussion can be introduced and explores barriers and facilitators to successful implementation.

Care providers in four studies reported that they quickly adapted to routine trauma discussion, and found it feasible within their workloads [47, 51–53].

*"I just think it's the initial getting used to. . . just even logging into it, and doing all of that was a hassle when I first started. It's "Oh, this is all so hard." But it's so simple now, because we're used to it. . . it's like anything,. . . any tool that you use over and over again, it becomes more simple"* [53].

Participants in seven studies discussed the challenges of ensuring all women are asked about prior trauma [46–48, 50–53]. Care providers taking part in studies by Choi and Seng (2014) and Stenson, Sidenvall and Heimer (2005) spoke of having preconceived notions of who might or might not have suffered trauma: *"If you have a feeling something isn't quite right, then it's easier. But just when you don't think. . ."* [52]. Participants in the study undertaken by Schmied et al. (2020) felt *"they're just not appropriate questions to ask"* [51], or avoided asking the question where a woman was felt to be 'keeping her distance'. Describing a successful implementation of routine trauma discussion, this participant in Reilly et al.'s study (2020) proposed that key individuals were helpful in getting more reluctant staff on board:

*"Even if you have people who are sceptical, if you are enthusiastic, and the clients get enthusiastic and really feel cared for, that automatically rolls over to the staff members that are sitting on the sidelines and saying "I don't know if all this is necessary". . . They start seeing that it is making a positive impact on people's lives. . ."* [50].

Participants in the study by Stenson, Sidenvall and Heimer (2005) [52] found it difficult to remember to raise the issue, resulting in some women not being asked about prior trauma. Documentation of abuse in hand-held notes could be a confidentiality risk, but several people taking part in the study suggested a check box in the records, to act as an aide memoire, and ensure the discussion is accorded the same importance as other issues [52]. Participants in Kohlhoff's study (2021) [48] proposed setting up a flag on the system to ensure effective information sharing among the maternity team.

Partner presence at appointments could influence the discussion of previous trauma [46, 48, 49, 52]. This was felt by some participants in the study by Stenson, Sidenvall and Heimer

(2005) to be a modern phenomenon, with one care provider proposing *"this generation of couples, the husband's there the whole time"* [52]. Whilst a participant in Kohlhoff et al.'s study (2021) [48] noted that *"pregnant women can be quite vulnerable and anxious, and it is good to have that support person with them"*, another pointed out:

> *"They're not going to be able to divulge anything while their partner's there, especially if their partner doesn't know about it, and sometimes that is the case"* [48].

Trauma discussions were more difficult for women with limited English [39, 52, 53]. Women often did not want to disclose sensitive issues through an interpreter, and where family members or partners were acting as interpreters, this provided a further barrier to disclosure. In the study by Stenson, Sidenvall and Heimer (2005) [52], staff reported that partners were often asked to sit in the waiting room while this discussion took place: few partners insisted on being present, but the midwives sometimes felt in a difficult position where the woman had limited English:

> *"I would really prefer a professional interpreter but on most occasions the men say 'no'. They want to do the interpreting"* [52].

### 'You go home and it's playing on your mind as you're cooking': The impact on care providers of hearing trauma disclosures

Hearing trauma disclosures could be challenging for clinicians [34, 46, 48, 52, 55]. Mollart, Newing and Foureur (2009) [55] explored this issue in detail in their Australian study in which they carried out focus groups with 18 midwives who undertook routine discussion of prior trauma. The midwife participants in the study reported that the cumulative, complex disclosures they heard affected them emotionally and impacted on their home and work life:

> *"You go home and it's playing on your mind as you're cooking. I don't know how long it usually goes on for, probably till you get that next bad case"* [55].

Some participants in the study reported that they continued to think about trauma disclosures after work:

> *"Sometimes I've gone home and actually worried about people, then you've got to remember that they told me this today and they've been living with this for how long? Just keep telling yourself that"* [55].

For some, this impacted on their family life:

> *"For me, I explode at home, I don't explode here [at work] because I know that no-one would put up with that kind of behaviour. But I do it to my kids, and that's not very good."* [55].

Marley et al. (2017) interviewed healthcare providers about their experiences of routine trauma discussion with Aboriginal women in Western Australia. A participant in the study expressed: *"At the end of the day, it's hard not to want to neck a bottle of wine to cope with [hearing their stories]"* [34]. The use of unhealthy coping strategies, such as excess alcohol, was echoed by a participant in Mollart, Newing and Foureur (2020):

"*I can debrief 10, 20, 30 times, and the information is still with me, and I don't know where to channel that sometimes. Sometimes you channel that into things that are probably not appropriate.*" [55].

Participants in studies undertaken by Marley et al. (2017) [34] and Mollart, Newing and Foureur (2009) [55] felt clinical supervision is vital for midwives carrying out routine trauma discussion, and those with no access to supervision expressed resentment. However, not all care providers who were offered supervision chose to participate in it:

"*The only way I know how to deal with it is I talk to colleagues. . . even though sometimes when you talk to colleagues, you know they're thinking about the booking-in they had, they're only half listening and you haven't actually been heard*" [55].

Stenson, Sidenvall and Heimer (2005) [52] was the only study to raise the issue of care providers who have experienced trauma themselves. The researchers carried out five focus groups with 21 midwives in Sweden to explore their views on discussing prior and current violence with women. The researchers noted that none of the participants spontaneously raised the issue. When the moderator brought it up '*several admitted that they had not considered the possibility that colleagues might have been subjected to abuse*" [52]. The group discussed whether midwives could act professionally in this situation and concluded that they could, but it might be difficult if they are still in a violent relationship. The group did not explore whether discussion of previous trauma might be more challenging for midwives with personal experience in this area, or how they could be supported.

## Discussion

This qualitative evidence synthesis found that although many women feel positive about routine discussion of previous trauma, this simple statement masks complexity. Women often find the conversation unexpected and intrusive; they expect their care to change after trauma disclosure and are disappointed if it does not; and women with limited English face additional challenges. The findings of the review suggest that many pregnant women are unaware of the extent of the trauma they have suffered, or its impact on their lives. This highlights the difficulty of discussing trauma with women who will be at very different stages of recovery.

The review underscores the importance of an established relationship and trust in trauma discussions. While midwife-led continuity models of care have been shown to be beneficial for mothers and babies, New Zealand is the only country to have achieved this at scale, and the national target of implementation of continuity of carer in the UK has now been removed [56–58]. Notwithstanding this, our review found that many women will not disclose previous trauma in the absence of a trusting relationship and will consequently fail to receive the care they need. Some women will choose not to disclose their histories; this is problematic because in the included studies, women who disclosed trauma were generally provided with information and follow-up care, while women who did not disclose trauma were not. Consideration should therefore be given to providing all women with information about the impact of trauma and the means to independently access support should they wish to do so.

Our review highlights specific areas of training which are necessary for maternity care professionals. The focus of trauma discussions should not be eliciting specific details of past trauma, but rather ascertaining what resources, support, and / or adjustments to the care plan would be helpful. It is important that women are forewarned of the discussion, including any limits to confidentiality and that disclosure of past trauma is entirely voluntary. The review

found that partner presence can be a barrier to trauma discussions. National Institute for Health and Care Excellence guidance (2021) [59] states that maternity care providers should ensure women are asked about domestic abuse in a private, one-to-one discussion; we propose that this also applies to discussion of previous trauma. For both women who have suffered trauma and professionals, support for routine discussion of previous trauma was found to be contingent on adequate time to explore these complex issues. This is a serious caveat in maternity systems which were already understaffed and have been further impacted by the COVID-19 pandemic, both in the UK and internationally [60]. Description of the discussion as easy and quick by some clinician participants in the included studies implies an unknowingly 'performative' approach to routine trauma discussion in which professionals feel they have given women the opportunity to disclose, but women may not feel comfortable disclosing or get the support they need.

This performative approach could also be a protective mechanism for clinicians who feel trauma discussions are emotionally difficult. Midwifery work carries a heavy emotional burden, with midwives suffering significantly higher levels of burnout, stress, anxiety, and depression than the general population [61–63]. Added to this, the review found that hearing women's histories of sexual assault, childhood abuse, and domestic violence can profoundly affect clinicians. None of the studies in the review explored whether discussion of previous trauma might be more challenging for midwives who have personally suffered trauma, or how they could be supported. There is a need for an appropriate support and supervision model which allows midwives the opportunity to explore the challenging situations they are exposed to. Reflective supervision would enable clinicians experiencing vicarious traumatisation or those whose own trauma experience have been triggered through their interactions with survivors to receive the support they need.

Women's perceptions and experiences of trauma discussion might be affected by characteristics such as class, ethnicity, or immigration status, but it was not possible to differentiate and draw distinct themes for different groups of participants. For example, the review found with high confidence that women are fearful of disclosing their histories in case it raises concerns that they are unable to safely care for their child: it may be that this is a greater barrier for women from population groups who experience disproportionately high levels of child welfare involvement [64]. Women with insecure immigration status are highly vulnerable to violence and abuse [65], but were excluded from almost all of the studies by default as their inclusion criteria required that participants could speak English: this is an area where further research is needed.

In 2014, the US Substance Abuse and Mental Health Services administration (SAMHSA) [66] published a seminal conceptual document about trauma-informed care, a model of service delivery which aims to develop an environment in which people who have experienced trauma feel safe and can build trust with their healthcare provider. This framework has been further developed and tailored to the perinatal period by others including Seng and Taylor (2015) [67], Sperlich et al. (2017) [68] and Law et al. (2021) [9]. SAMHSA (2014) [66] propose that trauma-informed approaches are underpinned by four assumptions: staff realise the prevalence of trauma and its impact on behaviour; they are able to recognise the signs of trauma, including by trauma screening; the organisation's policies and procedures take into consideration the experiences of trauma among service users and staff; and active steps are taken to resist re-traumatisation of clients and staff by ensuring psychological safety. Within a trauma-informed care framework, implementation of routine screening for previous trauma should be considered within the context of much broader changes to services, including staff training, continuity of carer, emotional support for staff, and service evaluation [9, 66–68]. Further research is needed to ascertain what resources and support are helpful to women who have

suffered trauma, bearing in mind that this must focus on the preferences and needs of the individual.

## Strengths and limitations of the review

This synthesis is the first to bring together the views of women and maternity care professionals on routine discussion of previous trauma in the perinatal period. We adopted an inclusive approach to the search, with broad search terms and multiple search strategies used to ensure no key articles were missed. The findings are strengthened by the large number of participants. We minimised the risk of over or under interpretation of the data through explicit positionality, reflexivity, and discussion.

However, the synthesis has limitations. All studies that met the inclusion criteria were from high-income countries, therefore the findings cannot be applied to women in low- or middle-income settings. Further, there were no studies set in Asia, and only in three in Europe. Research is needed on the optimal ways of addressing previous trauma in pregnancy and the perinatal period for different populations, in line with the Sustainable Development Goals and guidance by the European Parliament that sexual and reproductive health services should consider the needs of women who have suffered sexual or gender-based violence [69, 70].

## Conclusion

The review provides insight into the barriers and facilitators of women in high-income countries sharing their trauma histories, and clinicians asking about previous trauma. Areas for future research are highlighted. Discussions of previous trauma are complex and require careful consideration and sensitivity. While implementation of continuity of carer is no longer a target in maternity services in the UK, many women will not disclose previous trauma in the absence of a trusting relationship. As some women will not disclose their histories, consideration should be given to providing all women with information about the impact of trauma, and means of independently accessing support. These findings also highlight the need for time to undertake authentically sensitive and tailored discussions about trauma, and for appropriate support and supervision for care providers. There is a need for routine trauma discussions to be co-developed with childbearing women and the midwives who will be initiating the conversation, to ensure that women receive the support they need, and the wellbeing of care providers is protected.

## Supporting information

**S1 Checklist. PRISMA 2020 checklist.**
(DOCX)

## Acknowledgments

We would like to thank the EMPATHY study Research Collective for generously sharing their knowledge and expertise. Members of the collective are: Dr Laura Abbott, Juliet Albert, Kirsty Armstrong, Jill Benjoya Miller, Dr Emma Brooks, Dr Geraldine Butcher, Jo Doherty, Amber Jackson, Isobel Martin, Dr Elsa Montgomery, Sam Pointon, Sarah-Jayne Pomeroy, Erjola Sadria, Gill Skene, Memuna Sowe, Dr Kim Thomas, and Lucy Warwick-Guasp.

## Author Contributions

**Conceptualization:** Joanne Cull.

**Formal analysis:** Joanne Cull, Gill Thomson.

**Investigation:** Joanne Cull.

**Methodology:** Joanne Cull, Gill Thomson.

**Supervision:** Gill Thomson, Soo Downe, Michelle Fine, Anastasia Topalidou.

**Writing – original draft:** Joanne Cull.

**Writing – review & editing:** Joanne Cull, Gill Thomson, Soo Downe, Michelle Fine, Anastasia Topalidou.

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
