## [Decision Letter · Decision Letter 0]

23 Jan 2023

PONE-D-22-29747Views from women and maternity care professionals on routine discussion of previous trauma in the perinatal period: A qualitative evidence synthesisPLOS ONE

Dear Dr. Cull,

Thank you for submitting your manuscript to PLOS ONE. After careful consideration, we feel that it has merit but does not fully meet PLOS ONE’s publication criteria as it currently stands. Therefore, we invite you to submit a revised version of the manuscript that addresses the points raised during the review process.

We look forward to receiving your revised manuscript.

Kind regards,

Vidanka Vasilevski

Academic Editor

PLOS ONE

Journal Requirements:

Reviewers' comments:

Reviewer's Responses to Questions

**Comments to the Author**

1. Is the manuscript technically sound, and do the data support the conclusions?

Reviewer #1: Yes

Reviewer #2: Yes

2. Has the statistical analysis been performed appropriately and rigorously? 

Reviewer #1: N/A

Reviewer #2: N/A

3. Have the authors made all data underlying the findings in their manuscript fully available?

Reviewer #1: Yes

Reviewer #2: Yes

4. Is the manuscript presented in an intelligible fashion and written in standard English?

Reviewer #1: Yes

Reviewer #2: Yes

5. Review Comments to the Author

Reviewer #1: I appreciate the opportunity to review this well-written review paper on routine discussion of previous trauma in the perinatal period. The methodology is well articulated, and it is a strength that the authors have included a reflexive note for the authors. I offer the following queries and suggestions primarily centering on the results and discussion sections.

For the first theme of “‘I did not know how to say it, and no-one asked me’: should maternity care providers ask women about previous trauma?”, the authors note that although very many of the included studies’ participants found routine inquiry acceptable, that there are issues that are masking some complexities and contradictions. Some of these complexities are elucidated in this theme and other themes – notably the “‘You say it is confidential… but you are going to report me’: the importance of trust” theme. While I agree with the authors that routine inquiry would be better positioned in the context of a trusted relationship and with continuity of care, give that this is difficult to establish with the rise of the “hospitalist” system of care, it will be difficult to realize this ideal in the short term. Therefore, this underscores the need for more training on the part of perinatal professionals, in trauma-informed care, but also in trauma inquiry. The authors do acknowledge this in the discussion briefly, but I think it bears more discussion. Specifically, the authors might add that routine trauma inquiry should NOT be about eliciting specific details of past trauma, and should rather be present-focused in that clinicians should be communicating that they are trying to ascertain what the present needs are and how they might be able to assist the patient in the here and now with helpful resources and/or adjustments to the care plan that take the survivor’s specific needs into consideration. I also might add that there should be a preamble prior to inquiry that a) normalizes that you are asking all patients about this, b) that sharing such information is entirely voluntary, c) that any limits to confidentiality are fully explained, and that d) the point of the inquiry should be as mentioned above, present-focused, and that e) should only be done when there is adequacy of time to do the inquiry in a not-rushed manner, and that f) resources and supports are readily at hand and offered to all patients regardless of disclosure (as the authors have already alluded to).

p. 26 – note that “homely” in American English means “unattractive.” Perhaps use “homey” or some other word?

The theme title: ““Heavens! I forgot it!”: challenges to embedding trauma discussion in routine practice” could be better explained in the description of the theme. Also, I am not sure this quote is emblematic of the overall theme category – since this is where not only workload issues are covered but also where the authors are discussing partner presence at appointments as well as challenges related to language interpretation. Also, there are strategies that those working in IPV contexts employ to elicit trauma exposures despite the presence of the partner – perhaps include more information about this here or in the discussion?

I am glad that the authors also bring up the role of supervision – which should go hand in hand with training and support for clinicians. I feel this could be highlighted even more, especially the need for reflective supervision which would allow support for clinicians who experience vicarious traumatization or triggers of their own trauma experiences through their interactions with survivor clients.

Finally, the authors rightly call for systems change and the need for trauma-informed approaches. I feel this could also be augmented in the paper, perhaps by alluding not only to the Law et al 2021 guide but also to other published works regarding TIC in the perinatal period across provider contexts. I think there is opportunity here to provide more specificity as to how TIC might foster improvement in routine inquiry process, especially regarding the systems change the authors are calling for – what is really needed is a trauma informed continuum of responsivity, of which inquiry is one piece (See Sperlich et al., 2017 below, as well as some other likely relevant resources):

Kuehn, B. M. (2020). Trauma-informed care may ease patient fear, clinician burnout. JAMA, 323(7), 595-597.

Kuzma, E. K., Pardee, M., & Morgan, A. (2020). Implementing patient-centered trauma-informed care for the perinatal nurse. The Journal of Perinatal & Neonatal Nursing, 34(4), E23-E31.

Mosley, E. A., & Lanning, R. K. (2020). Evidence and guidelines for trauma-informed doula care. Midwifery, 83, 102643.

Nagle-Yang, S., Sachdeva, J., Zhao, L. X., Shenai, N., Shirvani, N., Worley, L. L., ... & Byatt, N. (2022). Trauma-Informed Care for Obstetric and Gynecologic Settings. Maternal and Child Health Journal, 1-8.

Seng, J. (2015). Trauma informed care in the perinatal period. Dunedin Academic Press Ltd.

Sperlich, M., Seng, J. S., Li, Y., Taylor, J., & Bradbury‐Jones, C. (2017). Integrating trauma‐informed care into maternity care practice: conceptual and practical issues. Journal of Midwifery & Women's Health, 62(6), 661-672.

Ward, L. G. (2020). Trauma-informed perinatal healthcare for survivors of sexual violence. The Journal of Perinatal & Neonatal Nursing, 34(3), 199-202.

Reviewer #2: This paper reports a qualitative evidence synthesis on routine discussion of previous trauma in the perinatal period and considers views from both women and maternity care professionals, which is a strength of the review. This is an important issue that is relevant to an international audience. The paper is well written, and a sensitive approach is evident. Overall, a thorough, systematic review is described, which incorporates a reflexive note regarding pre-existing beliefs of the authors on routine trauma discussion. The authors recognise the strengths and limitations of their review. I recommend that it is accepted subject to some minor revisions as detailed below.

Some further clarity would be helpful on the following points:

• P11, first 2 paragraphs – ‘To understand how best to support this work’ an explicit statement about which work would be helpful. As written, the statement implies that routine trauma conversations are needed, and the aim supports this assumption (‘how best to support women and professionals in having routine trauma conversations’). However, the review question, set out in the paragraph below, suggests that women and maternity care professionals will not necessarily be in favour of routine discussion, and this is something the review will investigate. I am concerned that these paragraphs could be interpreted as bias by the authors, which is not evident in the rest of the paper.

• P13, more information on search terms would be helpful. The authors indicate that they used broad search terms, but the only terms mentioned are ‘trauma-informed’ and ‘trauma informed’. Were any others used?

• P14, 4th line from bottom of page – ‘This enabled comparison between papers and reviewers.’ I found this comment confusing here and think it should be deleted as an explanation of comparison between papers and reviewers is provided on the following page

• P27 ‘a participant in Olsen…’, P32 ‘Participants in Marley et al. [34] and Mollart…’ Please could reference to these three studies be amended to say, ‘in the study/studies by’. This feels more respectful and is a construct used elsewhere.

• P27-28 ‘women reported feeling that there was no connection between trauma history and current pregnancy’. Please could this statement be qualified. As written, this could be misinterpreted as suggesting that all women in these 7 studies reported there was no connection between their trauma history and current wellbeing, which is not the case. Sometimes it is a small minority of women who hold this view.

• P30-31 ‘Heavens! I forgot it!’ This section feels very brief. The introductory sentence indicates that the theme will investigate how trauma discussion can be introduced and include practical steps to aid implementation. I do not recognise this description in the theme. Rather than indicating how trauma discussion can be introduced, it just indicates feasibility. It raises a couple of potential challenges but does not seem to include practical steps to implementation. This section would benefit from review by the authors.

• P32, the last theme is also very brief. I am wondering if there is any more relevant data in the papers that could be added to the section. If not, it would be helpful if the lack of data on this important issue could be addressed in the discussion.

• P35, ‘Further research is needed to ascertain what support is helpful to women who disclose trauma’. I would find a further exploration of this statement helpful. As presented, it implies that it will be possible to identify specific support for women who disclose trauma. I suspect that this will be dependent on individual women and how far through their journey to recovery they are.

• I have some concerns about the idea of ‘routine’ in relation to this work, as in the title and throughout the paper. It belies the complexity of working with people who have experienced trauma and the need for responsive care

Typographical errors

• P14 Line 3 of section Appraisal of the methodological quality of included studies: 1 mark was allocated to each question if the criteria was met. Should be ‘criterion’

• When in-text citations include mention of authors, there is a lack of consistency as to whether dates are included. Please could this be addressed.

• Table 1, country column – Sometimes the authors refer to United States and sometimes to America. Consistency would be helpful

6. PLOS authors have the option to publish the peer review history of their article (what does this mean?). If published, this will include your full peer review and any attached files.

Reviewer #1: No

Reviewer #2: **Yes: **Dr Elsa Montgomery

---

## [Author Response · Author response to Decision Letter 0]

6 Feb 2023

Please see 'response to reviewers' document attached.

---

## [Editor Report · Decision Letter 1]

27 Mar 2023

Views from women and maternity care professionals on routine discussion of previous trauma in the perinatal period: A qualitative evidence synthesis

PONE-D-22-29747R1

Dear Dr. Cull,

We’re pleased to inform you that your manuscript has been judged scientifically suitable for publication and will be formally accepted for publication once it meets all outstanding technical requirements.

Kind regards,

Vidanka Vasilevski

Academic Editor

PLOS ONE

---

## [Editor Report · Acceptance letter]

30 Mar 2023

PONE-D-22-29747R1 

Views from women and maternity care professionals on routine discussion of previous trauma in the perinatal period: A qualitative evidence synthesis 

Dear Dr. Cull:

I'm pleased to inform you that your manuscript has been deemed suitable for publication in PLOS ONE. Congratulations! Your manuscript is now with our production department. 

Kind regards, 

on behalf of

Dr. Vidanka Vasilevski 

Academic Editor

PLOS ONE